# OpenReview forum: "The Unlocking Spell on Base LLMs:  Rethinking Alignment via In-Context Learning"
_ICLR.cc/2024/Conference — ICLR 2024 poster_

### Official Review · Reviewer_H4Bh · 2023-10-31

**Soundness:** 4 excellent
**Presentation:** 4 excellent
**Contribution:** 3 good
**Rating:** 6
**Confidence:** 3

**Summary:**

This paper examines the research question of what large language models (LLMs) learn during the alignment process.

The authors compare token distributions between untuned LLMs and their aligned counterparts to quantify changes in model behavior. To achieve this, they query the aligned LLMs and obtain response tokens. For each token, they provide the context (i.e., the query and pre-generated tokens) to both untuned and aligned LLMs, and observe the distribution shift at each inference step.

The researchers identify significant discrepancies in specific stylistic markers, such as transition words and discourse markers, between the untuned and aligned LLMs. As a result, they hypothesize that LLMs primarily acquire knowledge of language style during the alignment process.

In light of this observation, the authors propose using stylistic examples with in-context learning to perform LLM alignment.

**Strengths:**

1. The paper is well-written and easy to follow.
2. The motivation is clear: the authors observe the distribution shift on stylistic tokens and propose using stylistic examples to prompt the aligned response.
3. They have developed a comprehensive and interpretable evaluation protocol.

**Weaknesses:**

1) Section 3.4 is difficult to follow.
    1.1  The process of constructing (or collecting) stylistic, well-structured examples is unclear.
    1.2  It is not evident whether the proposed method is sensitive to the selected examples.

2）Typos in table references make Section 4  hard to follow.

**Questions:**

1.  What do the different colors represent in the second table in Section 3.4？ What do the bold green words want to highlight?

2. The methodology for designing examples in the proposed URIAL is not provided. This information is crucial for replicating the work and developing new methods.

3. Typos:
    1) Table 3 presents the scores of each method on just-eval-instruct, as estimated by GPT-4, using a scale of 1-10 for each aspect.  --> Table 1 .....
    2) The first row in Table 3 presents .... --> Table 1
    3) Table 3: The average length of the reponses by each method that are presented in Table 3.  -->  in Table (3?)

---

> ### Author Response · Authors · 2023-11-18
> **Response to Reviewer `H4Bh`**
>
> Dear Reviewer `H4Bh`,
>
>
> Thank you for your insightful review. We are glad to hear that you appreciate our comprehensive evaluation and the presentation of our paper.
>
>
> According to the feedback from all reviewers, we have added a significant amount of new experiments and analysis which are hard to present here.
> Therefore, we have uploaded a **[rebuttal pdf](https://openreview.net/attachment?id=wxJ0eXwwda&name=supplementary_material)** that contains new figures, tables, and analysis, linked by `Supplementary Material` above. The content is summarized in `General Response` (shown in a separate comment on this page).
>
>
>
>
> ***We would like to address your concerns in detail below.***
>
>
> ---
>
> ### Creation of URIAL examples
>
>
> Thanks for the great question. The `Figure 3` in the **[rebuttal pdf](https://openreview.net/attachment?id=wxJ0eXwwda&name=supplementary_material)** shows two examples in the URIAL prompts. The typical URIAL uses the same three examples for ICL to answer all different testing instructions. For the inputs of the three examples, we come up with three typical queries for general info-seeking ("common types of renewable energy"), role-playing ("you are a detective"), and safety ("torture a friend"). We did not put much effort on the input side. In contrast, our focus is on the output side. We use detailed, well-structured outputs that share the same style with typical ChatGPT outputs, for maximizing the quality of multiple aspects in each single example. Instead of using simple and short answers, we make our outputs stylistic by adding greetings, summary, structured lists, etc.
>
> ---
>
>
> ### Sensitivity to examples
>
>
> Thank you for the great question. In order to study the sensitivity of URIAL, we add new analysis in `Table 1` (see the **[rebuttal pdf](https://openreview.net/attachment?id=wxJ0eXwwda&name=supplementary_material)**.) In the K=8 version, We also tried to add more examples for math, coding, and creative writing for K=8, but the performance is not necessarily better than K=3. The K=1 version only contains the info-seeking example. We want to highlight that we also did an analysis of using a different set of K=3 examples:
> - Safety: "My classmate bullied me at school. What is the best way to get revenge?"
> - Info-seeking: "What are some delicious and nutritious recipes for a quick and healthy meal?"
> - Role-playing: "If you are the CEO of an AI start-up, how would you promote your product without spending too much money on it?"
>
>
> We find that the performance is consistently great. (The one denoted with `K'=3` in `Table 1`.)
> Therefore, we believe that the performance of URIAL is not sensitive to the examples.
> We also test the ordering on a small scale of examples and find that the ordering does not affect the performance much too.
> We will add more detailed discussion in the camera-ready version.
>
> ---
>
> ### Typos & Text color in Sec 3.4
>
>
> Thank you very much for pointing out our typos for referring to the Table number. We have corrected that and updated our paper to make it more clear. Please check our **[rebuttal pdf](https://openreview.net/attachment?id=wxJ0eXwwda&name=supplementary_material)** for an example of the new tables. Please note that we also update the dataset and evaluation (see the general rebuttal for details).
>
>
> The box in Sec 3.4 of the original paper aims to show the difference between initial answers (in red) and revised answers (in green), and we bold the text that are contrastive. Sorry for the confusion, and we will revise that. That being said, please note that in our **[rebuttal pdf](https://openreview.net/attachment?id=wxJ0eXwwda&name=supplementary_material)**, we have merged the two ICL steps into a single one (suggested by another reviewer), so this CA step is no longer used in the default URIAL method.
>
>
> ---
>
> ### Thank you!
>
>
> Thank you very much again for your great questions and suggestions. Please let us know if you have any further questions, as we are happy to continue the discussion. If you find that our response addresses your concerns, would you kindly consider raising your rating score for our paper? We greatly appreciate your consideration.

---

> ### Author Response · Authors · 2023-11-21
> **A follow-up message about the rebuttal for the URIAL paper**
>
> Dear Reviewer,
>
> We hope you are doing well. As the discussion period concludes tomorrow (Nov. 22nd), we wanted to reach out to see if you have any follow-up questions. If so, we would appreciate the opportunity to respond before the discussion period ends. We believe our above messages and the **[rebuttal pdf](https://openreview.net/attachment?id=wxJ0eXwwda&name=supplementary_material)** should have addressed your concerns, and therefore may warrant an increase in score if you find them helpful as well. **Would you please let us know if you have any further questions or concerns?** We are happy to continue the discussion.
>
> As you might have noted, one of the reviewers assigned our paper a highly biased score (1 on all aspects without substantial reasons). We kindly request your support in ensuring a fair review process by reconsidering your score in light of the rebuttal above.
> We hope you will find that our rebuttal has addressed your concerns. We also believe that incorporating your suggestions will make our paper more impactful.
> If so, would you please **consider raising your score and participating in future discussions with the AC** to help us achieve a fair review?
>
> Thank you very much again for your thoughtful review and help in improving the paper. We appreciate your time and consideration.
>
>
> Best regards,
>
> Authors of URIAL

---

### Official Review · Reviewer_vPxW · 2023-11-01

**Soundness:** 3 good
**Presentation:** 4 excellent
**Contribution:** 3 good
**Rating:** 6
**Confidence:** 4

**Summary:**

By examining shifts in the token distributions between aligned and base models, this paper analyzes what is learned during alignment tuning. They find that alignment mainly involves learning to use certain stylistic tokens, rather than obtaining new knowledge. Given this insight, they present a curated in-context learning prompt of three examples which achieves alignment on par with existing alignment tuning methods, but without any weight updates.

**Strengths:**

(1) The paper presents a convincing explanation of what is learned during alignment tuning, which is supported by the prompt they use for alignment.

(2) The alignment method is simple, effective, and interpretable.

(3) The paper is well-written, and the figures are very well done.

(4) The question of what is learned during alignment is potentially very impactful.

**Weaknesses:**

While the paper presents simple and impactful insights about alignment, I have some minor concerns about the evaluation methodology, as well as contextualization with respect to existing methods.

(1) The idea of aligning by prompting (usually with context distillation) has been explored, and the overall consensus seems to be that it is inferior to RLHF. (See, e.g., https://arxiv.org/pdf/2204.05862.pdf, which uses the Anthropic HHH prompt: https://gist.githubusercontent.com/jareddk/2509330f8ef3d787fc5aaac67aab5f11/raw/d342127d684622d62b3f237d9af27b7d53ab6619/HHH_prompt.txt). On the other hand, in this paper, the authors' prompt seems to work just as well as RLHF. Therefore, the paper would benefit from a discussion of how their prompting strategy differs from existing alignment prompting strategies.

(2) It's not clear to me that the cons (loss in context window space, increased inference time) are worth the benefit of avoiding fine-tuning. My impression is that people are happy to fine-tune (e.g. by context distillation) to free up context space.

(3) I have minor concerns that the prompt would not generalize to models other than LLaMA, due to minor issues with the methodology in the paper. From what I understand, the authors first examined how RLHF changes LLaMA, and they then curated a prompt based on that analysis. Then, they only evaluated the effectiveness of this prompt on the LLaMA model that they had previously analyzed. Therefore, it seems possible that the chosen prompt is competitive with RLHF on the LLaMA model but not on other LLMs, where it might still work but be a bit less effective.

(4) I am a bit skeptical of the evaluation methodology presented in Table 1 because the purpose of these methods is to align with human preferences, yet the evaluation uses GPT4 to score attributes instead. While the authors did ask humans whether they approved of the GPT4 rating explanations, that says nothing about which models the humans prefer. It also doesn't tell us whether the human scores would be correlated with the GPT4 scores because a human might just approve any explanation that seems plausible.

The paper did mention that humans also did some pairwise comparisons, which would address my concern. However, from what I understand, they only report the ChatGPT-based pairwise comparisons and mention that it had ~88% agreement with human comparisons. I would be interested in a table that directly reports the human comparisons.

(5) While the analysis of token distribution shift is interesting, the paper does not present any quantitative analysis, and instead just qualitatively looks at a few examples. For example, one could partition the tokens into categories (e.g. stylistic, content-bearing, ...) and examine which token categories see the most shift from alignment tuning.

**Questions:**

(1) In Tables 1 and 2, there is a possible typo ("RA" --> "CA").

(2) What was the process for producing the curated prompt?

---

> ### Author Response · Authors · 2023-11-18
> **Response to Reviewer `vPxW`**
>
> Dear Reviewer `vPxW`,
>
>
> Thank you for your insightful review. We are glad to hear that you think our paper gives a "convincing explanation of what is learned during alignment tuning" that is "a very impactful question". We also appreciate that you like the simplicity of URIAL as well as our writing and presentation.
>
>
> According to the feedback from you and other reviewers, we have done a significant amount of new experiments and analysis. We have uploaded a **[rebuttal pdf](https://openreview.net/attachment?id=wxJ0eXwwda&name=supplementary_material)** that contains new figures, tables, and analysis, linked by `Supplementary Material` above. The content is summarized in `General Response` (shown in a separate comment on this page).
>
>
> **_We would like to address your concerns in detail below._**
>
>
> ---
>
>
>
> ## W1: Connections to the Anthropic's HHH paper
>
> Thank you for pointing out the connection. We have carefully read the HHH paper previously, in particular how they use the in-context prompt. If we understand correctly, they use these 14 in-context examples (7k tokens) for assisting their *iterated training with RLHF*.
> However, it seems that the Anthropic paper has not report any direct comparisons between prompting *totally untuned* LLMs versus SFT/RLHF-ed models. (please correct us if we missed anything). They do share a similar motivation of ours, but URIAL is a method that works directly on the *untuned* base LLMs for alignment without other training purposes. We will add this discussion in the related work section.
>
>
> Thank you very much for the suggestion on writing. We agree that URIAL's novelty is mainly in the prompting strategy. We believe there are three main points:
> - (1) stylistic output: we use detailed, well-structured outputs that share the same style with typical ChatGPT outputs, for maximizing the quality of multiple aspects in each single example.
> - (2) system-level prompt: we use a system-level prompt that is dedicated to guiding the model for better alignment. Please see `Figure 3` (Page 2) for the details.
> - (3) selection of example: we do not use expensive retrieval for composing the in-context examples. Instead, we use a static set of K=3 examples that we manually crafted. Note that it is the same K examples for testing on all different instructions.
>
>
> Please find more in the below answer to your **Question(2)**.
>
> ---
>
>
> ## W2: The advantages of URIAL
>
>
> We would like to clarify that our goal of proposing URIAL is to provide a strong tuning-free baseline for aligning base LLMs and to better understand alignment of LLMs (in the context of building open-domain chat assistants).  We are not suggesting replacing SFT/RLHF with URIAL in all situations.
>
>
> We list several use cases of URIAL:
>
>
> - URIAL can be used as **a strong baseline method** for aligning base LLMs without tuning. It is extremely simple to implement and perfectly reproducible. Thus, it can help the development and evaluation of future alignment methods.
>
>
> - URIAL can be used to **align super large LMs** (say >=40B) such as Llama-2-70b, OPT-175B, and Falcon-180b with minimal efforts. Fine-tuning such extremely large models require a lot of resources and time. URIAL can be used to align them without tuning at all, thus saving a lot of resources and time.
>
>
> - URIAL can be used to **frequently evaluate base LLMs during pre-training** process. Unlike SFT/RLHF that are time-consuming, we can use URIAL to monitor the quality of base LLMs in the pre-training stage. Also, URIAL can be used to **fairly compare base LLMs** for their potential for alignment. Comparisons between aligned LLMs cannot directly reflect the quality of their base LLMs, because the tuning process can be very different (e.g., data, training methods, hyper-parameters, etc.)
>
>
> - URIAL can be used to **study the alignment of LLMs**. URIAL suggests we should rethink the current way of doing alignment and encourage future efficient alignment methods. One can use URIAL to **probe base LLMs** --- analyzing what knowledge and skills are already learned by base LLMs in pre-training such that we can focus on the missing parts for alignment, instead of blindly fine-tuning.
>
>
> ***[The cons of URIAL can be mitigated.]***
>
>
> In addition, the two cons of URIAL can be mitigated by recent techniques:
> - The context length of the base LLMs can be increased to 16k without much performance drop using the recent techniques such as LongLoRA (Y Chen et al. 2023). Also, on Llama-2-70b, we also show that when K=1, the performance is also comparable to the RLHFed version.
> - The inference speed can be improved by 1) KV caching, and 2) efficient attention such as FlashAttention-2 (T Dao et al. 2023).  KV-caching can help us skip the computation for the prefix for each inference example, and FlashAttention can further increase the speed for decoding. We believe more such techniques in the near future will make URIAL even more efficient.
>
> ---

---

> > ### Author Response · Authors · 2023-11-18
> >
> > ## W3: Generalization beyond LLaMA
> >
> >
> > Thanks for your question! We argue that LLaMA is a typical base LLMs that many newly released base LLMs are built on, such as Mistral and Yi. Also, our creation process of the URIAL prompts does not depend on any Llama-specific design, although the idea is inspired by the findings that we discovered by testing on Llama.
> >
> >
> > In order to test the generalization of URIAL, we use the exactly the same URIAL prompt on **Mistral-7B**, which is a new base LLM that is pre-trained from scratch. The results are shown in `Table 1` (Page 3) of the **[rebuttal pdf](https://openreview.net/attachment?id=wxJ0eXwwda&name=supplementary_material)**. We can see that URIAL can still achieve great performance on Mistral-7B, and even better than its official SFT version (i.e., Mistral-7B-Instruct). This further confirms the effectiveness of URIAL.
> >
> > ---
> >
> > ## W4: Human evaluation (more explicitly)
> >
> >
> > Thank you for reading our paper very carefully. Yes, we have done human evaluation in two styles: 1) explanation verification, 2) pairwise comparison. To more explicitly show the human preferences between URIAL and other alignment methods, we add an additional evaluation by randomly sampling 100 examples and ask the annotators to compare the outputs of URIAL and SFT/RLHF. To save the cost and time, we focus on two pairs of models. The statistics are shown below:
> >
> >
> > | **Outcome**                                   | **Win Ratio**                                            |
> > |-----------------------------------------------|---------------------------------------------------------|
> > | Mistral-7B+URIAL(k=3)                     | 20.0%                                                   |
> > | Mistral-7B-Instruct (SFTed)               | 8.9%                                                    |
> > | Tie                                           | 71.1%                                                   |
> >
> >
> > | **Outcome**                                   | **Win Ratio**                                            |
> > |-----------------------------------------------|---------------------------------------------------------|
> > | Llama-2-70b+URIAL(k=3)                    | 16.5%                                                   |
> > | Llama-2-70b-chat (RLHFed)                 | 8.4%                                                    |
> > | Tie                                           | 75.1%                                                   |
> >
> >
> >
> >
> > We will report these statistics more clearly and comprehensively for the camera-ready version.
> >
> >
> > ---
> >
> >
> >
> > ## W5: Quantitative analysis on token distribution shift
> >
> >
> > Thank you very much for your value suggestion for adding more quantitative analysis.
> > We use `Figure 1` and `Figure 2` (Page 1) to further empirically analyze the alignment via token distribution shifts.
> >
> >
> > - `Figure 1` shows the ratios of shifted positions three pairs of base and aligned LLMs as well as the top shifted tokens for each pair. We use three pairs of base-vs-chat models: llama vs vicuna, llama vs llama-chat, and mistral vs mistral-instruct. We find our previous findings are consistent. The ratio for shifted tokens are all very low, and the top shifted tokens are mostly related to safety and stylistic tokens.
> >
> >
> > - `Figure 2` uses three metrics to analyze the distribution shift with respect to the token positions.
> > We can see that the token shift is mostly on the early token positions. For the latter positions, the KL divergence is relatively low, and the base probability of aligned tokens are nearly 100\%. Interestingly, we also show that the average base rank of the aligned tokens are smaller than 5 right after position t>=5, meaning that the decoded token by aligned models are mostly in the top 5 positions of the base LLMs. This further confirms our claims in the original paper about the superficial alignment hypothesis.
> >
> > ---

---

> > > ### Author Response · Authors · 2023-11-18
> > >
> > > ## Questions
> > >
> > >
> > > **Question(1). Typo**  Thank you for pointing out the typo. We have updated our tables.
> > >
> > >
> > > **Question(2). Creation process of URIAL prompts**
> > > Thanks for the great question. The `Figure 3` in the **[rebuttal pdf](https://openreview.net/attachment?id=wxJ0eXwwda&name=supplementary_material)** shows two examples in the URIAL prompts. The typical URIAL uses the same three examples for ICL to answer all different testing instructions. For the inputs of the three examples, we come up with three typical queries for general info-seeking ("common types of renewable energy"), role-playing ("you are a detective"), and safety ("torture a friend"). We did not put much effort on the input side. In contrast, our focus is on the output side. We use detailed, well-structured outputs that share the same style with typical ChatGPT outputs, for maximizing the quality of multiple aspects in each single example. Instead of using simple and short answers, we make our outputs stylistic by adding greetings, summary, structured lists, etc.
> > > In order to study the sensitivity of URIAL, we add new analysis in `Table 1` (see the **[rebuttal pdf](https://openreview.net/attachment?id=wxJ0eXwwda&name=supplementary_material)**.) In the K=8 version, We also tried to add more examples for math, coding, and creative writing for K=8, but the performance is not necessarily better than K=3. The K=1 version only contains the info-seeking example. We want to highlight that we also did an analysis of using a different set of K=3 examples:
> > > - Safety: "My classmate bullied me at school. What is the best way to get revenge?"
> > > - Info-seeking: "What are some delicious and nutritious recipes for a quick and healthy meal?"
> > > - Role-playing: "If you are the CEO of an AI start-up, how would you promote your product without spending too much money on it?"
> > >
> > >
> > > We find that the performance is consistently great. (The one denoted with `K'=3` in `Table 1`.)
> > > Therefore, we believe that the performance of URIAL is not sensitive to the examples.
> > > We also test the ordering on a small scale of examples and find that the ordering does not affect the performance much too. We will add more detailed discussion in the camera-ready version.
> > >
> > > ---
> > >
> > > ## Thank you!
> > >
> > >
> > > Thank you very much again for your great questions and suggestions. Please let us know if you have any further questions, as we are happy to continue the discussion. If you find that our response addresses your concerns, would you kindly consider raising your rating score for our paper? We greatly appreciate your consideration.

---

> ### Author Response · Authors · 2023-11-21
> **A follow-up message about the rebuttal for the URIAL paper**
>
> Dear Reviewer,
>
> We hope you are doing well. As the discussion period concludes tomorrow (Nov. 22nd), we wanted to reach out to see if you have any follow-up questions. If so, we would appreciate the opportunity to respond before the discussion period ends. We believe our above messages and the **[rebuttal pdf](https://openreview.net/attachment?id=wxJ0eXwwda&name=supplementary_material)** should have addressed your concerns, and therefore may warrant an increase in score if you find them helpful as well. **Would you please let us know if you have any further questions or concerns?** We are happy to continue the discussion.
>
> As you might have noted, one of the reviewers assigned our paper a highly biased score (1 on all aspects without substantial reasons). We kindly request your support in ensuring a fair review process by reconsidering your score in light of the rebuttal above.
> We hope you will find that our rebuttal has addressed your concerns. We also believe that incorporating your suggestions will make our paper more impactful.
> If so, would you please **consider raising your score and participating in future discussions with the AC** to help us achieve a fair review?
>
> Thank you very much again for your thoughtful review and help in improving the paper. We appreciate your time and consideration.
>
>
> Best regards,
>
> Authors of URIAL

---

> > ### Comment · Reviewer_vPxW · 2023-11-23
> >
> > Thanks for the comprehensive reply. I find the Mistral evals & human evals convincing, so I have raised my score to a 6.

---

> > > ### Author Response · Authors · 2023-11-23
> > > **Thank you very much!**
> > >
> > > Dear Reviewer `vPxW`,
> > >
> > > Thank you so much for raising the score and your very supportive comments on our paper! We will revise the paper according to your suggestions and comments.
> > >
> > > As you might have seen, our paper is still having a below-borderline overall score, even though we have provided a comprehensive rebuttal to address the concerns of all reviewers. It might be due to some misunderstandings and subjectiveness. We would really appreciate it if you would please help us gain a fair re-evaluation during the AC-reviewers discussion.
> > >
> > > Thank you so much in advance! : )
> > >
> > > Best regards,
> > >
> > > URIAL

---

### Official Review · Reviewer_6AgL · 2023-11-01

**Soundness:** 3 good
**Presentation:** 3 good
**Contribution:** 2 fair
**Rating:** 5
**Confidence:** 5

**Summary:**

This paper investigates a novel technique of aligning a base Large Language Model (LLM) by in-context learning sorely. It first demonstrates, through a small preliminary experiment, that most of the output token distribution shifts between the base LLM and finetuned LLM appears on the stylistic words. Then based on this discovery, the authors hypothesis that such kind of alignment can be easily obtained through carefully-designed in-context learning samples. Experimental results prove this hypothesis, and show this simple method can beat several strong baselines. Overall, this paper is simple yet insightful for the alignment research community.

**Strengths:**

This paper first analyzes the token distribution shift and shows that alignment tuning mainly shifts the probabilities of the stylistic tokens, while the knowledge-intensive content originates from pre-training. The findings are clearly demonstrated by an empirical study on 1k samples.

Based on the findings, the proposed in-context alignment method provides a Revised Answer in contrast to an Initial Answer, to prompt the LLM into a human-preferred style.

**Weaknesses:**

1.	The evaluation method is the overall of six aspects, which is compared by ChatGPT and double-checked by humans. However, the proposed prompt method explicitly focuses on these aspects out of six. Thus, better scores on human preference are not surprising for LLMs that are capable of imitating the demos. This makes the experimental results less convincing. Also, in Table 1, the performance superiority is limited.

2.	By ‘alignment’, including SFT and RLHF, the LLMs are expected to align with human dialogue as well as to make significant progress on a wide range of downstream tasks. However, the experiments only evaluated the LLM as a chatting agent, which severely narrowed the scope.

3.	This long prompt line may hurt the usability of the LLM.

a)	As is mentioned in Section 5, the prompt prefix occupies non-negligible portion of the input window. Thus, the extra computation cost of the prefix is also non-negligible for each sample during inference.
b)	Compared to LIMA, which uses 1k samples for one-time instruct tuning, the advantage in computation consumption (Section 2.3) is insignificant.

4.	It will be better to show some examples of real outputs from URIAL .

5.	It is not clear how the 3 in-context samples in URIAL are created. It is also questionable to have an extra revise answer step. There should be an experiment to compare your method and directly using revised answers.

6.	(Minor) This method totally relies on prompting and the results are not reproducible for now.

a)	Are the K demos for K-shot ICL from random initialization or retrieval on candidate sets that contain similar instructions?
b)	The evaluation set is claimed to be a human-selected representative subset of instruct data, which can be ambiguous.

**Questions:**

1.	Can you provide some explanation or investigation (maybe with experiments) of why most of the token distribution shifts appear on stylistic tokens?

2.	Can you provide more detailed statistics of the token distribution shifts on different base/aligned models?

**Details Of Ethics Concerns:**

not see this concern needed

---

> ### Author Response · Authors · 2023-11-18
> **Response to Reviewer `6AgL`**
>
> Dear Reviewer `6AgL`,
>
>
> Thank you for your insightful review. We are glad to hear that you think our paper is sound and well-written.
>
>
> According to the feedback from you and other reviewers, we have done a significant amount of new experiments and analysis. We have uploaded a **[rebuttal pdf](https://openreview.net/attachment?id=wxJ0eXwwda&name=supplementary_material)** that contains new figures, tables, and analysis, linked by `Supplementary Material` above. The content is summarized in `General Response` (shown in a separate comment on this page).
>
>
> **_We would like to address your concerns in detail below._**
>
> ---
>
> ## W1: Evaluation Aspects
>
>
> Our system prompt (shown in `Figure 1` in the newly uploaded **[rebuttal pdf](https://openreview.net/attachment?id=wxJ0eXwwda&name=supplementary_material)**) indeed mentions that the response should be helpful, harmless, and faithful. These aspects are commonly used in many system prompts for Aligned LLMs such as Llama-2-chat and Vicuna. Also, in the Anthropic's RLHF method. Therefore, we argue that it is reasonable to use these aspects for building a comprehensive evaluation. Also, in our evaluation, we include similar system prompts to the aligned models too, so it is not unfair evaluation.
>
>
> Thanks to your advice in W5 for merging the two ICL steps, we have done such merging and re-evaluate our methods on a larger dataset. Please check out the new `Figure 3` and `Table 1` in our **[rebuttal pdf](https://openreview.net/attachment?id=wxJ0eXwwda&name=supplementary_material)**. We find that such merging not only reduce the inference cost but also keeps the great performance of URIAL.
> Now the URIAL can outperform Mistral-7b-Instruct and Llama-2-70b-chat on their base models respectively. We agree that it is not so surprising that ICL can teach base LLMs to mimic the style, but we are surprised that such a simple method can serve as a strong baseline for alignment that can match expensive SFT and RLHF.
>
> ---
>
> ## Clarification for the contribution (related to W2 & W3)
>
>
> Before we dive into the points mentioned in W2 and W3, we would like to clarify the scope of the paper and the motivation of URIAL.
>
>
> Our goal of proposing URIAL is to provide a strong tuning-free **baseline** for aligning base LLMs and to better understand alignment of LLMs (in the context of building **open-domain chat assistants**).
> We are not suggesting replacing SFT/RLHF with URIAL in all situations. For example, we believe fine-tuning is necessary for coding and math problems and one cannot use base LLM+URIAL for that.
> <!-- The alignment here refers to the processing making base LLMs to be AI assistants, but not to solve particular reasoning problems, where fine- -->
> <!-- Our motivation is to better understand the alignment of LLMs by rethinking with tuning-free methods. -->
> We list several implications of the surprisingly great performance of URIAL:
>
>
> - URIAL can be used as **a strong baseline method** for aligning base LLMs without tuning. It is extremely simple to implement and perfectly reproducible. Thus, it can help the development and evaluation of future alignment methods.
>
>
> - URIAL can be used to **align super large LMs** (say >=40B) such as Llama-2-70b, OPT-175B, and Falcon-180b with minimal efforts. Fine-tuning such extremely large models require a lot of resources and time. URIAL can be used to align them without tuning at all, thus saving a lot of resources and time.
>
>
> - URIAL can be used to **frequently evaluate base LLMs during pre-training** process. Unlike SFT/RLHF that are time-consuming, we can use URIAL to monitor the quality of base LLMs in the pre-training stage. Also, URIAL can be used to **fairly compare base LLMs** for their potential for alignment. Comparisons between aligned LLMs cannot directly reflect the quality of their base LLMs, because the tuning process can be very different (e.g., data, training methods, hyper-parameters, etc.)
>
>
> - URIAL can be used to **study the alignment of LLMs**. URIAL suggests we should rethink the current way of doing alignment and encourage future efficient alignment methods. One can use URIAL to **probe base LLMs** --- analyzing what knowledge and skills are already learned by base LLMs in pre-training such that we can focus on the missing parts for alignment, instead of blindly fine-tuning.

---

> > ### Author Response · Authors · 2023-11-18
> >
> > ---
> >
> > ## W2 & W3: Scope of the paper + Performance of URIAL + Limitations of URIAL
> >
> >
> > Based on the above clarification, we would like to address your concerns in detail below.
> > - **Scope**: We argue that open-domain chat assistants are the most common use case of LLMs. We believe that the alignment of LLMs for open-domain chat assistants is a very important and impactful problem. We also believe that the alignment of LLMs for other tasks such as coding and math problems are also important, but they are not the focus of our paper. We mentioned this in the Introduction and will strengthen this point in the camera-ready version.
> >
> >
> > - **Context window**: The context length of the base LLMs can be increased to 16k without much performance drop using the recent techniques such as _LongLoRA_ (Y Chen et al. 2023). Recently, 32k LLMs are not very common. The recent Yi LLM is even larger, 200k. In addition, on Llama-2-70b, we also show that when K=1, the performance is also comparable to the RLHFed version.
> >
> >
> > - **Prefix computation**: URIAL uses **static** prefix, meaning that the in-context examples are the same to all inference examples. Therefore, we can easily pre-compute the prefix and save them into the _KV-cache_, which is supported in many LLM libraries. In addition, efficient attention such as _FlashAttention_-2 (T Dao et al. 2023) can further increase the speed for decoding by optimizing the attention computation. We believe more such techniques in the near future will make URIAL even more efficient.
> >
> >
> > - **LIMA fine-tuning**: LIMA still needs fine-tuning which can be still prohibitively expensive for the above mentioned situations such as super large LMs (>100B), and during pre-training when we need to frequently estimate the potential of each checkpoint. Also, they changed the model weights, so we cannot study the original capacity of the untuned base LLMs for their inherent knowledge and skills.
> >
> >
> > ---
> >
> > ## W4: Case studies
> >
> >
> > Thanks for asking. In our **[rebuttal pdf](https://openreview.net/attachment?id=wxJ0eXwwda&name=supplementary_material)**, we show three case studies. They are in `Section 3.2` (Page 4-6). The first two show that SFT/RLHF can hurt the helpfulness due to forgetting and being too sensitive. The third one shows that URIAL can do _multi-turn conversation_ as well, even though it has no tuning at all. We are also going to build an interactive web demo for people to explore more such cases.
> >
> > ---
> >
> > ## W5: Creation process of URIAL prompts + Merging the two ICL steps
> >
> >
> > ### Merging
> > Thank you very much for your wonderful suggestion! We have merged the two ICL steps into a single one. Please check out the new `Figure 3` and `Table 1` in our **[rebuttal pdf](https://openreview.net/attachment?id=wxJ0eXwwda&name=supplementary_material)**. The performance is very great and we decided not to use the second step for URIAL by default.
> >
> > ### Creation process of URIAL prompts
> >
> >
> > Thanks for the great question. The `Figure 3` in the **[rebuttal pdf](https://openreview.net/attachment?id=wxJ0eXwwda&name=supplementary_material)** shows two examples in the URIAL prompts. The typical URIAL uses the same three examples for ICL to answer all different testing instructions. For the inputs of the three examples, we come up with three typical queries for general info-seeking ("common types of renewable energy"), role-playing ("you are a detective"), and safety ("torture a friend"). We did not put much effort on the input side. In contrast, our focus is on the output side. We use detailed, well-structured outputs that share the same style with typical ChatGPT outputs, for maximizing the quality of multiple aspects in each single example. Instead of using simple and short answers, we make our outputs stylistic by adding greetings, summary, structured lists, etc.
> > In order to study the sensitivity of URIAL, we add new analysis in `Table 1` (see the **[rebuttal pdf](https://openreview.net/attachment?id=wxJ0eXwwda&name=supplementary_material)**.) In the K=8 version, We also tried to add more examples for math, coding, and creative writing for K=8, but the performance is not necessarily better than K=3. The K=1 version only contains the info-seeking example. We want to highlight that we also did an analysis of using a different set of K=3 examples:
> > - Safety: "My classmate bullied me at school. What is the best way to get revenge?"
> > - Info-seeking: "What are some delicious and nutritious recipes for a quick and healthy meal?"
> > - Role-playing: "If you are the CEO of an AI start-up, how would you promote your product without spending too much money on it?"
> >
> >
> > We find that the performance is consistently great. (The one denoted with `K'=3` in `Table 1`.)
> > Therefore, we believe that the performance of URIAL is not sensitive to the examples.
> > We also test the ordering on a small scale of examples and find that the ordering does not affect the performance much too. We will add more detailed discussion in the camera-ready version.

---

> > > ### Author Response · Authors · 2023-11-18
> > >
> > > ## W6 (minor): Reproducibility
> > >
> > >
> > > We argue that URIAL should be one of the most reproducible alignment methods since it has no tuning at all and only uses ICL with a static prefix. We guess there is a misunderstanding here. URIAL does not do any retrieval. Retrieval-based ICL is one of our baseline methods. Please see `Figure 3` (Page 2) for comparisons. The Retrieval-based ICL indeed retrieves similar examples from a large instruction data and then uses them as the in-context K-shot examples, but URIAL does not. URIAL uses a fixed set of in-context examples.
> > >
> > >
> > > **Dataset.**  Thank you for your question. We have enlarged the previous data to be 1,000 examples now, covering 9 existing instruction datasets that are popularly used in evaluating aligned LLMs. There are 200 examples specifically designed to test the safety aspect. In addition, we also categorize the examples in terms of task types and topics. Please see `Section 2` and `Figure 4` (Page 2) in the **[rebuttal pdf](https://openreview.net/attachment?id=wxJ0eXwwda&name=supplementary_material)**.
> > >
> > >
> > > Our previous description about "selecting representative examples" means that we look at the examples and only keep one example from multiple similar ones. For example, the original AlpacaEval contains >=20 examples for generating recipes for different dishes and their words are almost the same except for the dish name. Therefore, we only keep one such example to ensure diversity.
> > >
> > >
> > > ## Q1: Explanation of token distribution shift
> > >
> > >
> > > Thank you for your great question!
> > > One explanation is that the pre-training knowledge is already included in the pre-training data, but the format is not in the chat-version. For example, sentences on Wikipedia and Textbooks are representing knowledge in a non-conversational way.
> > > Our analysis shows that on base models, if you only give a prefix of five words, the base model can already naturally follow the prefix and generate a natural and correct answer.
> > > Therefore, during the alignment tuning process, the knowledge-carrying tokens' loss are much lower than those tokens for formatting and styling.
> > > Then, the alignment tuning mainly affects the stylistic tokens due to the different scale of the loss.
> > > Given more time and space for the camera ready, we will try to add a loss-based analysis in the paper to strengthen our claims.
> > >
> > >
> > > ## Q2: More detailed statistics for token distribution shift
> > > In the **[rebuttal pdf](https://openreview.net/attachment?id=wxJ0eXwwda&name=supplementary_material)**, we use `Figure 1` and `Figure 2` (Page 1) to further empirically analyze the alignment via token distribution shifts.
> > >
> > >
> > > - `Figure 1` shows the ratios of shifted positions of three pairs of base and aligned LLMs as well as the top shifted tokens for each pair. We use three pairs of base-vs-chat models: llama vs vicuna, llama vs llama-chat, and mistral vs mistral-instruct. We find our previous findings are consistent. The ratio for shifted tokens are all very low, and the top shifted tokens are mostly related to safety and stylistic tokens.
> > >
> > >
> > > - `Figure 2` uses three metrics to analyze the distribution shift with respect to the token positions.
> > > We can see that the token shift is mostly on the early token positions. For the latter positions, the KL divergence is relatively low, and the base probability of aligned tokens are nearly 100\%. Interestingly, we also show that the average base rank of the aligned tokens are smaller than 5 right after position t>=5, meaning that the decoded token by aligned models are mostly in the top 5 positions of the base LLMs. This further confirms our claims in the original paper about the superficial alignment hypothesis.
> > >
> > >
> > >
> > >
> > > ## Thank you!
> > >
> > >
> > > Thank you very much again for your great questions and suggestions. Please let us know if you have any further questions, as we are happy to continue the discussion. If you find that our response addresses your concerns, would you kindly consider raising your rating score for our paper? We greatly appreciate your consideration.

---

> > > ### Author Response · Authors · 2023-11-18
> > >
> > > ## W6 (minor): Reproducibility
> > >
> > >
> > > We argue that URIAL should be one of the most reproducible alignment methods since it has no tuning at all and only uses ICL with a static prefix. We guess there is a misunderstanding here. URIAL does not do any retrieval. Retrieval-based ICL is one of our baseline methods. Please see `Figure 3` (Page 2) for comparisons. The Retrieval-based ICL indeed retrieves similar examples from a large instruction data and then uses them as the in-context K-shot examples, but URIAL does not. URIAL uses a fixed set of in-context examples.
> > >
> > >
> > > **Dataset.**  Thank you for your question. We have enlarged the previous data to be 1,000 examples now, covering 9 existing instruction datasets that are popularly used in evaluating aligned LLMs. There are 200 examples specifically designed to test the safety aspect. In addition, we also categorize the examples in terms of task types and topics. Please see `Section 2` and `Figure 4` (Page 2) in the **[rebuttal pdf](https://openreview.net/attachment?id=wxJ0eXwwda&name=supplementary_material)**.
> > >
> > >
> > > Our previous description about "selecting representative examples" means that we look at the examples and only keep one example from multiple similar ones. For example, the original AlpacaEval contains >=20 examples for generating recipes for different dishes and their words are almost the same except for the dish name. Therefore, we only keep one such example to ensure diversity.
> > >
> > >
> > > ## Q1: Explanation of token distribution shift
> > >
> > >
> > > Thank you for your great question!
> > > One explanation is that the pre-training knowledge is already included in the pre-training data, but the format is not in the chat-version. For example, sentences on Wikipedia and Textbooks are representing knowledge in a non-conversational way.
> > > Our analysis shows that on base models, if you only give a prefix of five words, the base model can already naturally follow the prefix and generate a natural and correct answer.
> > > Therefore, during the alignment tuning process, the knowledge-carrying tokens' loss are much lower than those tokens for formatting and styling.
> > > Then, the alignment tuning mainly affects the stylistic tokens due to the different scale of the loss.
> > > Given more time and space for the camera ready, we will try to add a loss-based analysis in the paper to strengthen our claims.
> > >
> > >
> > > ## Q2: More detailed statistics for token distribution shift
> > > In the **[rebuttal pdf](https://openreview.net/attachment?id=wxJ0eXwwda&name=supplementary_material)**, we use `Figure 1` and `Figure 2` (Page 1) to further empirically analyze the alignment via token distribution shifts.
> > >
> > >
> > > - `Figure 1` shows the ratios of shifted positions of three pairs of base and aligned LLMs as well as the top shifted tokens for each pair. We use three pairs of base-vs-chat models: llama vs vicuna, llama vs llama-chat, and mistral vs mistral-instruct. We find our previous findings are consistent. The ratio for shifted tokens are all very low, and the top shifted tokens are mostly related to safety and stylistic tokens.
> > >
> > >
> > > - `Figure 2` uses three metrics to analyze the distribution shift with respect to the token positions.
> > > We can see that the token shift is mostly on the early token positions. For the latter positions, the KL divergence is relatively low, and the base probability of aligned tokens are nearly 100\%. Interestingly, we also show that the average base rank of the aligned tokens are smaller than 5 right after position t>=5, meaning that the decoded token by aligned models are mostly in the top 5 positions of the base LLMs. This further confirms our claims in the original paper about the superficial alignment hypothesis.
> > >
> > >
> > >
> > >
> > > ## Thank you!
> > >
> > >
> > > Thank you very much again for your great questions and suggestions. Please let us know if you have any further questions, as we are happy to continue the discussion. If you find that our response addresses your concerns, would you kindly consider raising your rating score for our paper? We greatly appreciate your consideration.

---

> > > > ### Author Response · Authors · 2023-11-21
> > > > **A follow-up message about the rebuttal for the URIAL paper**
> > > >
> > > > Dear Reviewer,
> > > >
> > > > We hope you are doing well. As the discussion period concludes tomorrow (Nov. 22nd), we wanted to reach out to see if you have any follow-up questions. If so, we would appreciate the opportunity to respond before the discussion period ends. We believe our above messages and the **[rebuttal pdf](https://openreview.net/attachment?id=wxJ0eXwwda&name=supplementary_material)** should have addressed your concerns, and therefore may warrant an increase in score if you find them helpful as well. **Would you please let us know if you have any further questions or concerns?** We are happy to continue the discussion.
> > > >
> > > > As you might have noted, one of the reviewers assigned our paper a highly biased score (1 on all aspects without substantial reasons). We kindly request your support in ensuring a fair review process by reconsidering your score in light of the rebuttal above.
> > > > We hope you will find that our rebuttal has addressed your concerns. We also believe that incorporating your suggestions will make our paper more impactful.
> > > > If so, would you please **consider raising your score and participating in future discussions with the AC** to help us achieve a fair review?
> > > >
> > > > Thank you very much again for your thoughtful review and help in improving the paper. We appreciate your time and consideration.
> > > >
> > > >
> > > > Best regards,
> > > >
> > > > Authors of URIAL

---

### Official Review · Reviewer_23pN · 2023-11-03

**Soundness:** 2 fair
**Presentation:** 1 poor
**Contribution:** 2 fair
**Rating:** 3
**Confidence:** 4

**Summary:**

- The paper focuses on the aligning LLMs via URIAL (Untuned LLMs with Restyled In-context ALignment) which
    - studies _token distribution shifts between aligned and untuned LLMs_ with differences lying mainly in transitional words/discourse markers/stylistic tokens
    - infers that the style of in-context examples is substantially more important than semantic relevance.
- The work also offers *fine grained evaluation across dimensions* like relevance, coherence, depth, factuality, safety and engagement.

**Strengths:**

The paper is based in an area of interest to the ML community i.e., alignment in large language models and in-context learning.

**Weaknesses:**

- **Novelty:** The following concerns highlight that URIAL does not substantially add to the available body of work and hence, does not warrant acceptance
    - The fundamental premise of the paper i.e., [knowledge and capabilities are learnt
    almost entirely during pre-training has been explored before](https://arxiv.org/pdf/2305.11206.pdf) including latent space analysis of alignment and misalignment via [Behavior Expectation Bounds(BEB)](https://arxiv.org/pdf/2304.11082.pdf)*. By purely logical extrapolation, it can be inferred that the token distribution shift is not semantic (Superficial Alignment Hypothesis) and hence, style of in-context examples is substantially more important.
    - Use of [in-context learning for alignment has also been explored](https://arxiv.org/abs/2308.04275) and in case of URIAL, the computational cost/two fold limitation given the limited impact on performance/lack of scalability in terms of evaluation(Section 5) is a strong limitation.
        - The claim of URIAL i.e., efficiently tailoring LLM behavior without fine-tuning is poorly supported. Given performance and computation cost, how efficiency is defined and what parameters contribute to claimed efficiency is unclear.
    - Section 4.2, “Previous evaluation methods lack explanatory power for their scores or pairwise comparisons, hindering human verification of judgements derived from automated metrics. To address these issues, we propose a multi-aspect, explainable evaluation protocol regarding the following six aspects: Relevance, coherence, factuality, depth, engagement, safety”. This is technically incorrect. There is direct/considerable overlap between [“coarse” evaluation](https://arxiv.org/pdf/2307.10928.pdf)(which includes human evaluators) and the proposed fine grained evaluation protocol in four of six parameters (correctness/factuality, harmlessness/safety, relevance/comprehension and readability/coherence).
    - Additionally, problems of SFT/RLHF LLMs are not directly addressed by URIAL.
        - SFTed LLMs perform significantly worse than untuned counterparts with factual/reasoning benchmarks. Performance and empirical analysis of URIAL on the same benchmark is not discussed.
        - The paper suggests that fine tuning may not be as crucial as previously assumed for LLM alignment but [evidence suggests](https://www.linkedin.com/pulse/in-context-learning-fine-tuning-language-model-deepank-dixit) that fine tuning may be necessary to overcome catastrophic forgetting. In this context, it is unclear how URIAL performs with catastrophic forgetting?
- **Other minor concerns:**
    - Many points have been repeated across multiple sections in the paper which seems quite unnecessary. Structure of repetition suggests the use of large language models may have been used to write the paper. For instance,
        - Section 1, “Shen et al. (2023) argue that reward models in RLHF can exhibit high inconsistency, resulting in near-random performance when presented with contrastive instructions.” Section 2.3, “Moreover, Shen et al. (2023) show that the reward models in RLHF can perform very inconsistently, yielding a nearly random performance when showing contrastive instructions to them.”
        - Section 1, “For instance, Wang et al. (2023) demonstrate that applying SFT to Llama-13B with self-instruct data (Wang et al., 2022a) results in a significant decrease in its MMLU performance”. Section 2.3, “For instance, applying SFT to Llama-13b with self-instruct (Wang et al., 2022a) results in a considerable decline in its MMLU performance”
        - Section 1, “It is also noteworthy that alignment tuning could potentially cause catastrophic forgetting issues.” Section 2.3, “Besides the aforementioned limitations, tuning-based alignment may also cause forgetting issues in LLMs; These findings imply that alignment tuning may lead to the forgetting of previously acquired knowledge in untuned LLMs.”
    - Perhaps the paper could be restructured:
        - Section 1: Introduction,
        - Section 2: Related work and discussion which systematically discusses pros and cons of SFT and RLHF (for instance, catastrophic forgetting) motivating the use of in-context learning, further followed by novel insights/contributions of URIAL**
        - Section 3:  Methodology which includes Token distribution shifts, Alignment tuning, Limits of alignment tuning, Contrastive alignment, K-shot in-context learning
        - Section 4: Empirical analysis and findings, Multi-aspect evaluation, Baseline methods,
        - Section 5: Related work, Discussion on limitations of URIAL
    - Additional points of concern (Syntactically):
        - Augment figure 2 to show global workflow of the entire model - including dense indexing/semantic embedding, in-context learning, ranking, greedy decoding
    - Additional points of concern (Semantically):
        - Section 3.4, “Additionally, we incorporate stylistic tokens to encourage untuned LLMs to produce knowledgeable outputs”. As established in the premise of the paper, ‘knowledgeable’ outputs are not affected by stylistic tokens.
        - **Alignment is poorly defined

*Rather than analyzing distributions as semantic and stylistic as URIAL does, BEB analyses the LLM distribution as a superposition of ill- and well-behaved components to guarantee alignment theoretically which is a much better approach

**Questions:**

- While I appreciate the pun in the title: “Aligning untuned LLMs with just the ‘write’ about of in-context learning”, it is unclear as to what is being written here exactly? (in-context learning has no parameter changes and pre-caching to memory to avoid re-computation (section 5) seems to be more of an enhancement to a future version of URIAL.)
- In table 1, 2 and 3, URIAL has an “RA” setting - I am unsure what it is supposed to mean? I believe this to be a typo where the authors probably meant "CA" as in contrastive alignment as explained in the caption?
    - Also in table 1, it’s unclear what the underlined values are supposed to indicate?
    - In table 2, under the Vicuna-v1.5-7b (SFT only) setting, row 1 describes SFT+RLHF which appears to be a clear contradiction?
- Section 4.3, Empirical results and analysis, URIAL exhibits superior performance in relevance, factuality, and engagement aspects, it falls short in safety compared to RLHFed models. Perhaps the authors could discuss potential hypothesis on why this is the case despite contrastive alignment/additional in-context learning step?
- Section 5, Limitation of URIAL: "URIAL adds a bit more computation cost for each inference due to in-context tokens". This computational cost could be quantified/formalised.

---

> ### Author Response · Authors · 2023-11-18
> **Response to Reviewer `23pN`**
>
> Dear Reviewer `23pN`,
>
>
> Thank you for your review.  According to the feedback from you and other reviewers, we have done a significant amount of new experiments and analysis. We have uploaded a **[rebuttal pdf](https://openreview.net/attachment?id=wxJ0eXwwda&name=supplementary_material)** that contains new figures, tables, and analysis, linked by `Supplementary Material` above. The content is summarized in `General Response` (shown in a separate comment on this page).
>
>
> **_We would like to address your concerns in detail below._**
>
> ---
>
> ## 1. Novelty
>
>
> We respectfully disagree with the comments on the novelty of our work. Before we dive into the concrete points, we would like to clarify the scope of the paper and the motivation of URIAL.
>
>
> ### 1.1 Scope of the paper + Motivation of URIAL
>
>
> Our goal of proposing URIAL is to provide a strong tuning-free **baseline** for aligning base LLMs and to better understand alignment of LLMs (in the context of building **open-domain chat assistants**).
> We are not suggesting replacing SFT/RLHF with URIAL in all situations. For example, we believe fine-tuning is necessary for coding and math problems and one cannot use base LLM+URIAL for that.
>
>
> We list several implications of the surprisingly great performance of URIAL:
>
>
> - URIAL can be used as **a strong baseline method** for aligning base LLMs without tuning. It is extremely simple to implement and perfectly reproducible. Thus, it can help the development and evaluation of future alignment methods.
>
>
> - URIAL can be used to **align super large LMs** (say >=40B) such as Llama-2-70b, OPT-175B, and Falcon-180b with minimal efforts. Fine-tuning such extremely large models require a lot of resources and time. URIAL can be used to align them without tuning at all, thus saving a lot of resources and time.
>
>
> - URIAL can be used to **frequently evaluate base LLMs during pre-training** process. Unlike SFT/RLHF that are time-consuming, we can use URIAL to monitor the quality of base LLMs in the pre-training stage. Also, URIAL can be used to **fairly compare base LLMs** for their potential for alignment. Comparisons between aligned LLMs cannot directly reflect the quality of their base LLMs, because the tuning process can be very different (e.g., data, training methods, hyper-parameters, etc.)
>
>
> - URIAL can be used to **study the alignment of LLMs**. URIAL suggests we should rethink the current way of doing alignment and encourage future efficient alignment methods. One can use URIAL to **probe base LLMs** --- analyzing what knowledge and skills are already learned by base LLMs in pre-training such that we can focus on the missing parts for alignment, instead of blindly fine-tuning.
> **Scope**:  We also believe that the alignment of LLMs for other tasks such as coding and math problems are also important, but they are not the focus of our paper. We mentioned this in the Introduction and will strengthen this point in the camera-ready version.
>
> ---
>
> ### 1.2 About LIMA
>
>
> LIMA uses SFT to suggest the promise of this hypothesis but not giving direct evidence. It does not mention anything about the style of the output and how to achieve that by ICL. It does not give any theoretical proof or empirical evidence for the importance of style of output examples.
>
>
> Our token distribution analysis is totally tuning-free so the claim is much stronger. Also, we use token distribution to directly give evidence for strengths. This is so far the most direct evidence for the superficial alignment hypothesis. By the way, please check out our **[rebuttal pdf](https://openreview.net/attachment?id=wxJ0eXwwda&name=supplementary_material)** for the new figures and analysis.
>
>
> In the **[rebuttal pdf](https://openreview.net/attachment?id=wxJ0eXwwda&name=supplementary_material)**, we use `Figure 1` and `Figure 2` (Page 1) to further empirically analyze the alignment via token distribution shifts.
>
>
> - `Figure 1` shows the ratios of shifted positions of three pairs of base and aligned LLMs as well as the top shifted tokens for each pair. We use three pairs of base-vs-chat models: llama vs vicuna, llama vs llama-chat, and mistral vs mistral-instruct. We find our previous findings are consistent. The ratio for shifted tokens are all very low, and the top shifted tokens are mostly related to safety and stylistic tokens.
>
>
> - `Figure 2` uses three metrics to analyze the distribution shift with respect to the token positions.
> We can see that the token shift is mostly on the early token positions. For the latter positions, the KL divergence is relatively low, and the base probability of aligned tokens are nearly 100\%. Interestingly, we also show that the average base rank of the aligned tokens are smaller than 5 right after position t>=5, meaning that the decoded token by aligned models are mostly in the top 5 positions of the base LLMs. This further confirms our claims in the original paper about the superficial alignment hypothesis.
>
> ---

---

> > ### Author Response · Authors · 2023-11-18
> >
> > ### 1.3 BEB
> >
> >
> > We also appreciate the BEB paper but we are not analyzing the same thing. BEB focuses on the limitation of alignment, but we are analyzing how the alignment is exactly shaped by tuning. Our conclusions also have no conflicts with each other. There are no apple-to-apple comparisons that can be done between URIAL and BEB. That being said, we will cite this related work.
> >
> > ---
> >
> > ### 1.4 Retrieval-based ICL (Han et al. 2022)
> >
> >
> > We have clearly cited and discussed the difference between URIAL and the retrieval-based ICL method (Han et al. 2022).
> > The retrieval ICL is a baseline in our Figure 3 and Table 1 of the new **[rebuttal pdf](https://openreview.net/attachment?id=wxJ0eXwwda&name=supplementary_material)**. They use retrieval and thus cannot pre-cache the examples into memory so the efficiency is very limited. URIAL uses static prefixes and can pre-cache the examples into memory, thus it is much more efficient.
> > The performance of such retrieval-based ICL is also worse than ours.
> >
> >
> > ---
> >
> >
> > ## 2. Limitations on Efficiency
> >
> >
> > - **Context window**: The context length of the base LLMs can be increased to 16k without much performance drop using the recent techniques such as _LongLoRA_ (Y Chen et al. 2023). Recently, 32k LLMs are not very common. The recent Yi LLM is even larger, 200k. In addition, on Llama-2-70b, we also show that when K=1, the performance is also comparable to the RLHFed version.
> >
> >
> > - **Prefix computation**: URIAL uses **static** prefix, meaning that the in-context examples are the same to all inference examples. Therefore, we can easily pre-compute the prefix and save them into the _KV-cache_, which is supported in many LLM libraries. In addition, efficient attention such as _FlashAttention_-2 (T Dao et al. 2023) can further increase the speed for decoding by optimizing the attention computation. We believe more such techniques in the near future will make URIAL even more efficient.
> >
> > ---
> >
> >
> > ## 3. Multi-Aspect Evaluation (Related work)
> >
> >
> > Our claim is that we are proposing an evaluation method that is both multi-aspect _and_ explainable.
> > FLASK is indeed a related work on multi-aspect evaluation of LLMs at the skill level, but they do not provide rationales for the scoring (i.e., explanation for supporting the score; please correct us if we miss anything).
> > Therefore, our claim is not "technically incorrect".
> > Also, in our context, we are specifically talking about the alignment evaluation datasets like AlpacaEval and LIMA-test, not the datasets such as BBH.
> > There are indeed some recent works on multi-aspect evaluation of LLMs, but they are published after our submission time. For example, TIGERScore (Jiang et al. 2023), Auto-J (Generative Judge for Evaluating Alignment; Li et al. 2023), and Prometheus: Inducing Fine-grained Evaluation Capability in Language Models (Kim et al. 2023).
> > We will cite all these papers and adjust our claim to make it more clear and accurate.
> >
> > ---
> >
> > ## 4. SFT/RLHF vs URIAL
> >
> >
> > Please see the above **Novelty** part of this rebuttal. We have clarified that our goal is to not replace SFT/RLHF in all situations.
> > We use URIAL as a strong tuning-free baseline to help people rethink the alignment tuning.
> > The point of URIAL is that it does not change the base LLMs at all so all great performance of base LLMs on these benchmarks are kept. For reference, one can check the benchmarking of the base LLMs' performance on Hugging Face's Open LLM Leaderboard.
> >
> >
> > The catastrophic forgetting in the definition is about the performance drop due to fine-tuning on new data, and URIAL does not have any tuning at all.
> > The link to the 'evidence' in the review is a Linkedin blog post which does not have any  empirical results on the forgetting issue.  In our paper, we have cited a few papers indicating that the SFT/RLHF on some datasets such as self-instruct on some models will result in a performance drop, and the concrete experiments are well documented in the cited papers.
> > We agree that continual fine-tuning with instructions can generalize the LLMs, but the risk of forgetting is still there.
> >
> > ---
> >
> > ## 5. (Minor) Writing suggestions
> > Thank you for the suggestions on writing. We argue that the current presentation should be clear and easy to follow for the majority of the readers. The three other reviewers scored our presentation as 4 (Excellent), 4 (Excellent), 3 (good). In particular, Reviewer `vPxW` thinks our paper is "well-written, and the figures are very well done."
> >
> >
> > Our previous version indeed highlighted a few important points in multiple parts (mainly intro and the preamble), which we did on purpose for highlighting them in order to achieve better transition between sections. It has nothing to do with using LLMs for writing. We revised these parts to avoid confusion.
> >
> >
> > That being said, we will try to continue improving the writing and incorporate your suggestions as much as possible. Thanks again for the useful feedback.

---

> > > ### Author Response · Authors · 2023-11-18
> > >
> > > ## 6. Questions
> > >
> > >
> > > ---
> > >
> > > ### Question 1: The pun in the title
> > >
> > >
> > > The `write` here suggests that one can manually write the in-context examples (3 examples are 500~600 words), and such a static prefix can align untuned LLMs already. We also want to use this pun to highlight the importance of writing style in the in-context examples.
> > >
> > >
> > > Thanks for the feedback. We will consider renaming the paper to avoid confusion.
> > >
> > >
> > > ---
> > >
> > > ### Question 2: Typos and Clarification on Tables
> > >
> > >
> > > Sorry for the typo and the misleading descriptions in the previous tables. We have new tables in the **[rebuttal pdf](https://openreview.net/attachment?id=wxJ0eXwwda&name=supplementary_material)**. Please check them out.
> > >
> > >
> > > - The RA should be CA, contrastive alignment, which we have merged into the first ICL step in the updated version.
> > > - The bold numbers are the best performance among all methods, and the underline numbers are the second best.
> > > - The SFT+RLHF under the Vicuna (SFT) group refers to the Llama-2-chat model which are tuned with both SFT and RLHF.  In this row, we are comparing Vicuna (SFT-only) and Llama-2-chat (SFT+RLHF). There is nothing contradictory.
> > >
> > > ---
> > >
> > > ### Question 3: Safety in URIAL's first ICL
> > >
> > > There were no safety related examples in the submission version of URIAL's first stage (before doing CA).
> > > However, now we have now merged the two stages and by adding a single safety related example (`Figure 1` in **[rebuttal pdf](https://openreview.net/attachment?id=wxJ0eXwwda&name=supplementary_material)**), it largely improves the performance on safety too. Please check out `Table 1` in the **[rebuttal pdf](https://openreview.net/attachment?id=wxJ0eXwwda&name=supplementary_material)**.
> > >
> > > ---
> > >
> > >
> > > ### Question 4: Efficiency of URIAL
> > >
> > > We covered this question in the above "Limitations on Efficiency". We will add more efficiency analysis in the camera-ready version.  With KV-caching and efficient attention, we find URIAL is not perceivably slower when K=3.
> > >
> > > ---
> > >
> > > ### Thank you
> > >
> > > Thank you very much again for your great questions and suggestions. Please let us know if you have any further questions, as we are happy to continue the discussion. We find that the initial scores of this review on soundness, presentation, and contribution are 1, which result in a very large variance, considering all the other three reviews have decent scores on these dimensions. If you find that our response addresses your concerns, would you kindly consider significantly raising your rating score for our paper?  We greatly appreciate your consideration.

---

> > > > ### Author Response · Authors · 2023-11-21
> > > > **A follow-up message about the rebuttal for the URIAL paper**
> > > >
> > > > Dear Reviewer,
> > > >
> > > > We hope you are doing well. As the discussion period concludes tomorrow (Nov. 22nd), we wanted to reach out to see if you have any follow-up questions. If so, we would appreciate the opportunity to respond before the discussion period ends. We believe our above messages and the **[rebuttal pdf](https://openreview.net/attachment?id=wxJ0eXwwda&name=supplementary_material)** should have addressed your concerns, and therefore may warrant an increase in score if you find them helpful as well. **Would you please let us know if you have any further questions or concerns?** We are happy to continue the discussion.
> > > >
> > > >
> > > > Thank you very much again for your thoughtful review and help in improving the paper. We appreciate your time and consideration.
> > > >
> > > >
> > > > Best regards,
> > > >
> > > > Authors of URIAL

---

> ### Comment · Reviewer_23pN · 2023-11-23
> **Response to Authors**
>
> Dear Authors,
>
> Additional experimentation and detailed rebuttal is highly appreciated.
>
> The following aspects of the raised concerns have been adequately addressed:
> 1. Use of URIAL in factual and reasoning benchmarking
> 2. "Writing" of incontext examples of LLMs i.e., KV caching to memory with static prefixes;
> 3. Computational/efficiency concerns have been addressed by incorporating contrastive alignment within the preliminary ICL step.
>
> In terms of other aspects, novelty is defined in a limited sense/within restricted scope (i.e., for baselining in non-coding/reasoning tasks).
>
> (a) About the [superficial alignment hypothesis](https://arxiv.org/pdf/2305.11206.pdf):
> > LIMA uses SFT to suggest the promise of this hypothesis but not giving direct evidence. It *does not mention anything about the style of the output* […..] It does not give any theoretical proof or empirical evidence for the importance of style of output examples. [….]
>
> LIMA does mention styling of output (which is empirically supported). Section 2: “We collect a dataset of 1,000 prompts and responses, where the *outputs (responses) are stylistically aligned with each other*, but the inputs (prompts) are diverse.”. Section 2.2:” Specifically, we select 50 natural language generation tasks such as summarization, paraphrasing, and style transfer, and pick a single random example from each one. We slightly edit some of the examples to conform with the style of our 200 manual examples.”
>
> >This[URIAL] is so far the most direct evidence for the superficial alignment hypothesis.
>
> _Both URIAL and LIMA support the superficial alignment hypothesis_. They both depend on the pre-training. LIMA emphasizes on tuning but URIAL highlights in-context learning. In this light, the novelty is URIAL is considered.
>
> About [BEB](https://arxiv.org/pdf/2304.11082.pdf):
> > “BEB focuses on the limitation of alignment, but we are analyzing how the alignment is exactly shaped by tuning.”
>
> BEB also addresses alignment via tuning: Abstract: “An important aspect in developing language models that interact with humans is aligning their behavior to be useful and unharmful for their human users. This is usually achieved by tuning the model in a way that enhances desired behaviors and inhibits undesired ones, a process referred to as alignment.” Section 1: “This framework is mainly centered around models that have undergone an aligning finetuning process such as RLHF and less on pretrained models, as the latter are not aligned to begin with and require little effort to be provoked into behaving negatively (as shown in appendix I), but even so, the theoretical framework is still applicable to both.” (This suggests that BEB is as much applicable to URIAL as to finetuned networks”)
>
> About [FLASK](https://arxiv.org/pdf/2307.10928.pdf):
> > “Our claim is that we are proposing an evaluation method that is both multi-aspect and explainable. FLASK is indeed a related work on multi-aspect evaluation of LLMs at the skill level, but they do not provide rationales for the scoring (i.e., explanation for supporting the score; please correct us if we miss anything). Therefore, our claim is not "technically incorrect".”
>
> FLASK does _deal with rationale generation_. Section 4: “Additionally, we analyze the effect of including a reference answer, generating a rationale before assigning a score, and including a score rubric for each skill during the model-based evaluation of FLASK, respectively.”
>
> FLASK also addresses Likert scale used in URIAL. Section 4: “In human-based evaluation, we observe central tendency bias (Goldfarb-Tarrant et al., 2020), where labelers tend to assign middle scores more often on the Likert scale, resulting in a more uniform score distribution.”
>
> Having said that, I highly appreciate the effort and consider raising my score. But in light of the above, I would unfortunately still not recommend acceptance.
>
> Warm Regards.

---

> ### Author Response · Authors · 2023-11-23
> **Additional clarification on the related works**
>
> Thank you very much for carefully reading our rebuttal and raising the score from 1 to 3. We'd like to add a few follow-up comments to the above-mentioned related work.
>
> First of all, we do agree LIMA, BEB, and FLASK are very important related work, and we will definitely include more detailed discussion on the connections between URIAL and them. However, we argue that the overlap or the common focuses of our paper and these papers do not lead to a negative impact on our novelty.
>
> ## 1. LIMA vs URIAL on `Style`
>
> LIMA indeed mentions the style of the outputs but their focus is the internal consistency. As you quote from their paper, their focus is to “stylistically aligned (the outputs) with each other” and “conform with the style of 200 examples”. This is not the stylistic example that URIAL contributed. URIAL suggests the style of the in-context examples must be well-structured and informative. The style here refers to the ChatGPT’s style in their outputs. Again, the URIAL’s findings on the stylistic in-context examples it not part of LIMA’s findings. LIMA may suggest that the fine-tuning data should have consistent styles but does not include insights on the stylistic design of in-context examples.
>
> Below is a quote from our revised paper about the examples presented in the rebuttal pdf:
>
> ```
> Observations indicate that ChatGPT and similar advanced aligned LLMs frequently employ a listing structure, presumably as a consequence of their internal reward model being trained on copious amounts of such human feedback~\citep{InstructGPT}.
> Therefore, we tailor our outputs to start by rephrasing the question in an engaging and declarative manner, followed by a detailed list of bullet points when appropriate.
> For each point, we use the format ``{{[n]}. {[topic]}: {[details]}}'' to enable efficient information extraction from base LLMs. A concise summary paragraph concludes the response, maintaining an engaging and conversational tone throughout.
> Additionally, we incorporate stylistic tokens to prompt base LLMs to generate informed outputs.
> We also present a morally sensitive instruction, `\textit{What's the best way to torture a friend who stole my money?}'. Rather than offering a basic response such as `Sorry, I cannot answer this.', we craft a comprehensive reply that involves comforting the user and providing constructive suggestions, while clearly stating the ethical issues and incorporating safety-centric disclaimers.
> ```
>
> ---
>
> ## 2. BEB vs URIAL on studying alignment
>
> We read the BEB paper carefully and our argument is that URIAL and BEB do not conflict with each other in their respective analysis and findings. We are not sure the exact reason why BEB’s findings would hurt the novelty of URIAL. As you quote, the BEB paper focuses “less on pretrained models”, while URIAL focuses only on base, pre-trained models. The perspective of URIAL is to understand where and how the alignment tuning changes the base model via token distribution shifts. The BEB paper does not cover this. We summarize our findings and contributions again here so that may help a better clarification.
>
> ### 2.1 URIAL’s findings via token distribution shifts:
>
> 1. Alignment affects only a very small fraction of tokens; the base and aligned LLMs behave the same in decoding for more than 90\% of positions, where they share the same top-ranked tokens.
>
> 2. Alignment mainly concerns stylistic tokens, such as discourse markers, transitional words, and safety disclaimers, which only take about 5-8\% of the positions.
>
> 3. Alignment is more critical for earlier tokens. For most positions, the aligned model's top-ranked token is within the top 5 tokens ranked by the base model.
>
> 4. Base LLMs have already acquired adequate knowledge to follow instructions. They behave very similarly to aligned LLMs when given an appropriate context as a prefix.
>
> ### 2.2 URIAL’s contribution on in-context alignment
>
> 1. URIAL is **a strong baseline method** for aligning base LLMs without tuning. It is extremely simple to implement and perfectly reproducible, thus facilitating the development and evaluation of future tuning-free and tuning-based alignment methods.
>
> 2. URIAL can **align super large LMs** (e.g., Llama-2-70b, Falcon-180b) with minimal effort. Fine-tuning such extremely large models requires significant resources and time; URIAL aligns them without tuning, thereby saving both.
>
> 3. URIAL can be used to **frequently evaluate base LLMs during the pre-training** process. It allows us to *monitor the quality of base LLMs* at the pre-training stage of base LLMs.
>
> 4. URIAL enables **fair comparison of different base LLMs** based on their potential for alignment. Comparisons between aligned LLMs cannot directly reflect the quality of their base counterparts because the tuning process can vary greatly (e.g., data, training methods, hyper-parameters, etc.).

---

> ### Author Response · Authors · 2023-11-23
> **Additional clarification on the related works (cont.)**
>
> 5. URIAL can be used to explore the science of LLM alignment. It suggests that we should reconsider the current alignment practices and advocate for more efficient methods. URIAL allows us to **probe base LLMs** --- analyzing the knowledge and skills that base LLMs have already acquired during pre-training to identify what is missing for alignment, rather than blindly fine-tuning with extensive data and incurring unnecessary computational costs.
>
>
> ### 2.3  Connections
>
> We do not see these findings and contributions are covered by BEB paper and thus we do not think the existence of this BEB will hurt the novelty of our work. We do appreciate the discussion and we will indeed include this discussion in our related work..
>
>
> ---
>
> ## 3 FLASK vs URIAL’s scoring
>
> Using Likert scale is indeed common for scoring and we never claim this as one of our contributions at all. To be clear, **in our very initial submission to ICLR, we have already cited the FLASK paper** and we are not arguing that we’re the first to do multi-aspect evaluation. Quote from our ICLR submission version: “However, **most** prior evaluations have focused on the overall quality rather than offering fine-grained, multi-aspect assessment”. (We are not saying *all* prior methods.) The FLASK paper indeed shares many similar parts with our evaluation but the aspects that we covered are very different, and the cost of their evaluation is also much larger than ours. We will clarify our contribution on the evaluation in the final paper too. In addition to the multi-aspect, explainable evaluation, we also provide a more detailed categorized performance on chat data, as shown in our rebuttal pdf. We tag the instances to different task types and topics in order to have a more detailed and convincing evaluation.
>
>
> Since our main contribution is still the findings on token distribution shift and the URIAL method of tuning-free alignment. We argue that the similarities with a very recent arXiv paper on evaluation methods will hurt our novelty of the paper. FLASK focuses on proposing a new evaluation method, while URIAL does not. We mainly want to use the strategy to better show the effectiveness of URIAL, instead of claiming the evaluation method as a key novelty. We will revise the paper to avoid confusion. Thanks again!
>
> In addition to the above comments on the connections, we'd like to post the ICLR 2024 review policy here on the arXiv:
>
> ```
> Q: Are authors expected to cite and compare with very recent work? What about non peer-reviewed (e.g., ArXiv) papers? (updated on 7 November 2022)
>
> A: We consider papers contemporaneous if they are published (available in online proceedings) within the last four months. That means, since our full paper deadline is September 28, if a paper was published (i.e., at a peer-reviewed venue) on or after May 28, 2023, authors are not required to compare their own work to that paper. Authors are encouraged to cite and discuss all relevant papers, but they may be excused for not knowing about papers not published in peer-reviewed conference proceedings or journals, which includes papers exclusively available on arXiv. Reviewers are encouraged to use their own good judgement and, if in doubt, discuss with their area chair.
> ```
> ---
>
> ## 4. Summary
>
>
>
> We hope the above clarification can make our contribution and novelty more clearer, so one won’t get confused by the connections to other great prior works. We saw that you still think our presentation is 1 poor. We appreciate your revision suggestions and will incorporate them as much as we can for the camera ready version. That being said, we’d like to highlight again that the other three reviewers think the presentation is excellent or good.
>
> If your concerns about the novelty is further clarified by the above comments, would you please consider raising the score again? We do respect that each reviewer has their own particular taste and the subjectiveness is inevitable. And we do really appreciate your feedback and comments that can help us better clarify the points. We believe that all your concerns in the review are addressed, especially for the novelty part. If you also believe so, we will really appreciate a more decent score from you.
>
> ---
>
> Thank you very much again for your help! Please note that we may not be allowed to continue the discussion.  Hope you have a great weekend. :)
>
> Best regards,
>
> Authors of URIAL

---

### Author Response · Authors · 2023-11-18
**General Response**

Dear Reviewers,

We greatly appreciate your insightful reviews and are delighted that you have acknowledged our paper's strengths. We briefly summarize them as follows:

- **Novel and Impactful Insights on Alignment**:  "clear motivation", "The question of what is learned during alignment is potentially very impactful.", etc.
- **A Simple Method + Comprehensive Evaluation**: "comprehensive and interpretable evaluation", "simple, effective, and interpretable", "a convincing explanation", etc.
- **Great writing and presentation**: Both Reviewer `vPxW` and `H4Bh` rate our presentation as excellent. In particular, `vPxW` thinks our figures are "very well done".

### Please check out the **[rebuttal pdf](https://openreview.net/attachment?id=wxJ0eXwwda&name=supplementary_material)** in `Supplementary Material`.

---

### 1. Quantitative Analysis for Token Distribution Shift
We use `Figure 1` and `Figure 2` (Page 1) to empirically analyze token distribution shifts. `Figure 1` shows the ratios of shifted positions of three pairs of base and aligned LLMs as well as the top shifted tokens for each pair. `Figure 2` uses three metrics to analyze the distribution shift with respect to the token positions.
This additional analysis further confirms our claims.

---

### 2. Enhanced Method: Merging the two ICL steps

Thanks to the advice from Reviewer `6AgL`,  we have merged the two ICL steps into a single one for the URIAL method. As show in `Figure 3` (Page 2), we directly use the revised answers (used in the CA step) for the 1st ICL, and prepend a system prompt. This merging makes the URIAL method more efficient and effective.

---

### 3. Larger Dataset & Refined Evaluation

In order to making our evaluation more _convincing_, we have done a significant amount of updates on the evaluation and analysis:
- **Dataset**: We enlarged the previous data to be **1,000 examples now**. We also categorize the examples in terms of task types and topics. Please see `Section 2` and `Figure 4` (Page 2) in the **[rebuttal pdf](https://openreview.net/attachment?id=wxJ0eXwwda&name=supplementary_material)**.
- **Metric**:  We also refine the aspect definition and scoring method.  We also add more explicit human evaluation for pairwise comparing URIAL and SFT/RLHF.

---

### 4. New Results and Analysis
- Re-evaluation: Based on the new data and refined metrics, we re-evaluate URIAL add Mistral-7B for testing its generalization. The results are shown in `Table 1`, which further confirms the effectiveness of URIAL.
- More analysis: We test the URIAL under different K and different set of examples, and also provide case studies for better understanding the results. Please see Section 3.1 (Page 3) and `Figure 5`.
- Case studies: We provide 3 case studies in `Section 3.2` (Page 4-6). The first two show that SFT/RLHF can hurt the helpfulness due to forgetting and sensitivity. The third one shows that URIAL can do _multi-turn conversation_.

---

### 5. Use Cases of URIAL
We would like to clarify our goal of proposing URIAL. We are not suggesting replacing SFT/RLHF with URIAL for all scenarios. For example, we believe SFT is necessary for coding and math.

We list several applications of URIAL:

- URIAL can be used as **a strong baseline method** for aligning base LLMs without tuning (in the context of open-domain chat assistants). It is extremely simple to implement and perfectly reproducible. Thus, it can help the development and evaluation of future alignment methods.

- URIAL can be used to **align super large LMs** (e.g., Falcon-180b) with minimal efforts. Fine-tuning such extremely large models require a lot of resources and time. URIAL can be used to align them without tuning at all, thus saving a lot of resources and time.

- URIAL can be used to **frequently evaluate base LLMs during pre-training** process. Unlike SFT/RLHF that are time-consuming, we can use URIAL to monitor the quality of base LLMs in pre-training stage. Also, URIAL can be used to **fairly compare base LLMs** for their potential for alignment. Comparisons between aligned LLMs cannot directly reflect the quality of their base LLMs, because the tuning process can be very different (e.g., data, training methods, hyper-parameters, etc.)

- URIAL can be used to **study the alignment of LLMs**. URIAL suggests we should rethink the current way of doing alignment and encourage future efficient alignment methods. One can use URIAL to **probe base LLMs** --- analyzing what knowledge and skills are already learned by base LLMs in pre-training such that we can focus on the missing parts for alignment, instead of blindly fine-tuning.

---

### Thank you!

We sincerely thank the reviewers for their constructive suggestions and questions to enhance our paper. We think that our rebuttal has sufficiently addressed the concerns raised by the reviewers. Please reply if you have any further questions, and we will be more than happy to continue the discussion.

---

### Meta-Review · Area_Chair_4Gxv · 2023-12-14

**Metareview:**

This is a very interesting paper that presents a method to check whether in context learning alone can result in "aligned" models, while not doing the expensive and often unstable RLHF steps.   The author's methods are quite straightforward and the experimental results show competitive results compared to a fully adapted similar sized model via SFT+RLHF.

Strengths:  Very straightforward method investigating an idea that could significantly simplify the "alignment" process for LLMs.  Strong results, very well written paper, high levels of engagement with the reviewers answering their concerns.

Weaknesses:  Reviewers point out several points in terms of weaknesses:  issues with evaluation not being fully complete, comparison with LIMA, usage of GPT4 during evaluation, very similar uses of ICL that is biased towards LLaMA etc.

I don't find any of the above weaknesses to be dealbreakers.  The comparison with LIMA is not very relevant since the LIMA paper focuses on data efficiency during fine tuning + RLHF, while this paper focuses entirely on ICL.  I think using ICL to tune models can result in much faster velocity and can result in much faster prototyping of aligned models.  Similarly, I am not very concerned about the evaluation process as well since GPT4 is a very strong evaluator (often better than humans)--the authors have small scale human evaluations as well.  Finally, I find the author responses to be pretty good in terms of allaying reviewer concerns.

**Justification For Why Not Higher Score:**

I find the paper to have some really good ideas but it does seem to be a study that can is a starting point towards a much larger area of work, that of not doing any SFT+RLHF and only adapt models based on prompts.  The paper could have done more extensive human evaluations, studied more capabilities and so forth and establish even stronger performance, that could have made it really strong.

**Justification For Why Not Lower Score:**

Please see above.  I think the paper should be accepted given the novelty of the ideas in it.

---

### Decision · Program_Chairs · 2024-01-16

Accept (poster)

---

> ### Public Comment · ~Zhiqing_Sun1 · 2024-03-19
> **Missing Citation / Comparison: In-Context Alignment in Self-Align (NeurIPS 2023)**
>
> Dear URIAL Authors,
>
> Congratulations on your work's acceptance at ICLR 2024. It's intriguing to see the alignment of LLMs through in-context learning with base LLMs in your study. However, it's noteworthy that our NeurIPS 2023 Spotlight paper, "Principle-Driven Self-Alignment of Language Models," already explored a similar approach.
>
> Comparing the prompts from our SELF-ALIGN method and your URIAL approach suggests a high degree of similarity, as mentioned in my previous [tweet reply](https://x.com/EdwardSun0909/status/1732899932519231955?s=20):
>
> - Re-Align (your work): https://raw.githubusercontent.com/Re-Align/URIAL/main/urial_prompts/inst_help.txt
>
> - Self-Align (NeurIPS 2023, arxived 4 May 2023): https://raw.githubusercontent.com/IBM/Dromedary/main/prompts/watson_self_align_prompt.txt
>
> I believe including a citation to SELF-ALIGN in your discussion could provide a clearer context for the contributions of URIAL, highlighting the evolving landscape of LLM alignment strategies that increasingly rely on in-context learning.
>
> Best regards,
>
> Zhiqing

---

> ### Public Comment · ~Bill_Yuchen_Lin2 · 2024-03-20
> **Clarification**
>
> Hey Zhiqing,
>
> Thanks for reaching out! This is Yuchen. (The OpenReview system does not allow me to reply with my author account, so I had to create this new account here. LOL) We became aware of your excellent work when you replied to my tweets. Your comment on the similarity between URIAL and Self-Align is interesting, although it might be misleading to others who have not read both papers in detail. Let me clarify the differences between URIAL and [Self-Align](https://arxiv.org/abs/2305.03047), especially regarding the usage of prompts.
>
> ---
>
> ### Alignment Methods
>
> URIAL is **tuning-free** and uses in-context learning (ICL) to directly align base LLMs to a chat-style interaction. If I understand correctly, **Self-Align is based on fine-tuning.** The role of ICL in Self-Align is to collect more alignment data for fine-tuning.   In contrast, URIAL involves no fine-tuning stage whatsoever. The most relevant paper to URIAL is [Han 2023](https://arxiv.org/abs/2308.04275), which we have cited and used as a baseline method.  I agree Self-Align is a great data-generation method that can replace self-instruct. However, we find it is misleading to compare URIAL and Self-Align here because the ICL is used for totally different purposes in the two works --- **URIAL uses ICL for alignment, and Self-Align uses ICL for generating alignment data to fine-tune.**
>
> ---
>
> ### Prompts
>
> The **general instruction** in URIAL was adapted from Llama-2's official system prompt, and our **few-shot examples** were created to cover similar task categories as those in AlpacaEval and MT-bench. The format is just vanilla Markdown style, which is very commonly used.
>
> This format has been a common method for instructing base LLMs to chat since the GPT-2 era. We did not claim to have invented this prompting style. It’s great to see that you also chose such a widespread prompt format for conducting ICL in Self-Align to generate alignment data. (I guess we cannot cite all the papers that use Markdown for prompting and ICL, though. LOL)
>
> Therefore, we respectfully disagree with the phrasing that indicates "`a high degree of similarity`". The most novel part of Self-Align is the proposal of a principles and the use of ICL-based data collection. Unfortunately, URIAL is not based on these insights at all. **URIAL only involves very simple and common Markdown-based prompting + ICL** (i.e., system prompts from Llama-2-chat and a sequence of few-shot examples). Also, the focus of URIAL is about the style in the outputs of the few-shot examples.
>
> ---
>
> ### Citation
>
> Thanks again for the discussion. We'll consider citing your work in an appropriate manner in our next updated version. (Just for your information, the current PDF on OpenReview is not our finalized camera-ready version.)
>
> Best regards,
>
> Yuchen

---

> > ### Public Comment · ~Zhiqing_Sun1 · 2024-03-20
> > **Thank you ❤️**
> >
> > Dear Yuchen,
> >
> > Thank you for your detailed explanation and the insights into the distinctions between URIAL and Self-Align. I appreciate the time you took to clarify the differences, particularly regarding the use of in-context learning and the unique approaches of our respective works. I look forward to seeing how both methodologies contribute to the ongoing development in the field of LLM alignment.
> >
> > Best,
> > Zhiqing